# CD33 rs2455069 SNP: Correlation with Alzheimer’s Disease and Hypothesis of Functional Role

**DOI:** 10.3390/ijms23073629

**Published:** 2022-03-26

**Authors:** Fabiana Tortora, Antonella Rendina, Antonella Angiolillo, Alfonso Di Costanzo, Francesco Aniello, Aldo Donizetti, Ferdinando Febbraio, Emilia Vitale

**Affiliations:** 1Institute of Biochemistry and Cell Biology, National Research Council (CNR), Via Pietro Castellino 111, 80131 Naples, Italy; fabiana.tortora@ibbc.cnr.it (F.T.); antonella.rendina@ibbc.cnr.it (A.R.); ferdinando.febbraio@cnr.it (F.F.); 2Centre for Research and Training in Medicine of Aging, Department of Medicine and Health Science “V. Tiberio”, University of Molise, 86100 Campobasso, Italy; angiolillo@unimol.it (A.A.); alfonso.dicostanzo@unimol.it (A.D.C.); 3Department of Biology, University of Naples Federico II, 80126 Naples, Italy; faniello@unina.it

**Keywords:** CD33, Alzheimer’s disease, sialic acid, phagocytosis

## Abstract

The *CD33* gene encodes for a member of the sialic-acid-binding immunoglobulin-type lectin (Siglec) family, and is one of the top-ranked Alzheimer’s disease (AD) risk genes identified by genome-wide association studies (GWAS). Many *CD33* polymorphisms are associated with an increased risk of AD, but the function and potential mechanism of many CD33 single-nucleotide polymorphisms (SNPs) in promoting AD have yet to be elucidated. We recently identified the *CD33* SNP rs2455069-A>G (R69G) in a familial form of dementia. Here, we demonstrate an association between the G allele of the rs2455069 gene variant and the presence of AD in a cohort of 195 patients from southern Italy. We carried out in silico analysis of the 3D structures of *CD33* carrying the identified SNP to provide insights into its functional effect. Structural models of the CD33 variant carrying the R69G amino acid change were compared to the CD33 wild type, and used for the docking analysis using sialic acid as the ligand. Our analysis demonstrated that the CD33-R69G variant may bind sialic acid at additional binding sites compared to the wild type, thus potentially increasing its affinity/specificity for this molecule. Our results led to a new hypothesis of rs2455069-A>G SNP as a risk factor for AD, suggesting that a long-term cumulative effect of the CD33-R69G variant results from the binding of sialic acid, acting as an enhancer of the CD33 inhibitory effects on amyloid plaque degradation.

## 1. Introduction

According to recent reports, it is estimated that the number of cases of dementia in the world is expected to rise to 152 million by 2050 as the population ages [1,2], with AD being the most common form of dementia currently [3,4]. Most AD cases are sporadic, with late-onset AD (LOAD) occurring at the age of 65 years or older. Approximately 5% of cases are classified as early-onset AD (EOAD), as reported in a systematic review and meta-analysis [5], where the age of onset is before 65 years and a percentage is considered as familial [6,7]. Except for a small number of early-onset cases who are afflicted due to highly penetrant single-gene mutations, AD, especially in the late-onset form, is genetically heterogeneous, with a polygenic or oligogenic risk inheritance [7]. Genome-wide association studies (GWAS) identified several genetic loci associated with increased susceptibility to LOAD affecting lipid metabolism, tau binding proteins, amyloid precursor protein (APP) metabolism [8], protein catabolism, immune cells, and microglia [9]. Of these, microglial cells are a key participant in LOAD pathogenesis, with many susceptibility loci, including *CD33*, playing a role [10,11,12,13,14,15]. CD33 belongs to a subfamily of the Ig superfamily comprising a single N-terminal V-set Ig domain that binds sialic-acid-containing glycans through a conserved positively charged arginine that interacts with the negatively charged C-1 carboxyl group on the sialic acid [16]. CD33 may be activated continuously by sialic-acid-containing glycoproteins and glycolipids, which are structural elements of amyloid plaques in the brains of Alzheimer’s disease patients, resulting in an inhibition of microglia-mediated immune activation [17,18]. CD33 can affect microglial phagocytosis through crosstalk with the transmembrane receptor TREM2. Knockout of CD33 resulted in improved cognition and decreased amyloid β pathology in 5xFAD mice, whereas these effects were completely abrogated by additional knockout of TREM2 [19]. TREM2, through an interaction with DAP12/TYROBP, stimulates the tyrosine–protein kinase SYK, which in turn triggers the activation of microglial cell phagocytosis [20,21]. This activation could be counteracted by sialic-acid-activated CD33, which stimulates SHP1/SHP2 tyrosine phosphatases, thereby inhibiting microglial phagocytosis [13,21,22].

In accordance with the effect of *CD33* on the inhibition of microglial phagocytosis, primary microglial cells from *CD33* knockout mice [12] or deletion of *CD33* in human macrophages and microglia resulted in an increased phagocytosis of amyloid β1-42 [23]. This was further supported by targeting *CD33* in a mouse model of AD using an adeno-associated virus (AAV) vector encoding a *CD33* microRNA (miRCD33), which demonstrated a reduction in amyloid β40 and β42 levels in brain extracts [24].

Different *CD33* polymorphisms have been positively or negatively correlated with AD susceptibility. These SNPs can affect the expression level, the structure, and the function of CD33, leading to changes in microglia-mediated clearance of amyloid β, likely contributing to the accumulation of senile plaques in the brain [12,25,26,27]. The CD33 variant rs3865444(A) negatively correlates with AD, has a reduced *CD33* expression [12], and is coinherited with rs12459419(T), which modulates the splicing efficiency of the IgV domain exon involved in sialic acid binding [13,28].

We recently identified the SNP rs2455069 in exon 2 of the human *CD33* gene on chromosome 19, which is involved in an unusual familial form of dementia [29]. This SNP has been previously identified in a block of eight correlated SNPs in *CD33* significantly associated with cognitive decline in univariate analysis in the all-female cohort, with highly significant effects on hyaluronan synthase-1 (HAS1) expression in the temporal cortex [30]. The presence of the minor allele at rs2455069 leads to a change from an arginine to a glycine in the position 69 (R69G). Here, we demonstrate a positive correlation between the *CD33* SNP rs2455069 and Alzheimer’s disease in a cohort of patients from a region of southern Italy, and secondly, that the R69G amino acid change affects the CD33 structure in a way that alters its binding affinity for sialic acid residues. We propose that the mechanism of the long-term cumulative effects of the CD33-R69G variant are mediated through an increased binding of sialic acid, which then acts as a more effective enhancer of the CD33 inhibitory effects on amyloid plaque degradation.

## 2. Results

### 2.1. Genetic Analysis

Based on the evidence of the possible involvement in an unusual form of dementia of the SNP rs2455069-A>G in exon 2 of *CD33* gene [29], leading to a change from arginine to glycine in position 69, we widened the analysis of this SNP to a cohort of AD patients from southern Italy. We found that the genotype frequency for the rs2455069 polymorphism in the control group was 11.1% (*n* = 15) for the GG, 53.0% (*n* = 77) for the AG, and 31.8% (*n* = 43) for the AA genotypes. In AD patients, the frequency of GG, AG, and AA genotypes were 17.9% (*n* = 35), 65.1% (*n* = 127), and 16.9% (*n* = 33), respectively. While genotypes in the LOAD sample were not in Hardy–Weinberg equilibrium (HWE, *p* = 0.00013), the genotype frequencies in controls did not deviate significantly from the Hardy–Weinberg equilibrium (HWE) (*p* = 0.08301). The G allele frequency was 39.6% in controls and 50.5% in AD patients (*p* = 0.00582). A significant association between AD and the GG or AG genotype was found (Table 1), particular in the dominant model (AG + GG vs. AA), leading to the hypothesis that the R69G change may have had an impact on the increased risk of LOAD in the analyzed cohort.

### 2.2. In Silico Structural Analysis

To evaluate the impact of the R69G change on the CD33 protein function, we carried out a series of in silico analyses. Six human CD33 partial structures (Siglec-3) were available in the RCSB Protein Data Bank [31] (Appendix A). We prepared the monomeric form of each CD33 structure, summarized in Appendix A, using the web-based platform CHARMM-GUI. To reduce possible artifacts due to binding with mimetic substrates, we used a protein model based on the CD33 structure PDB-ID 5IHB, representing the protein region interacting with NAG as a nonspecific ligand. We compared the two structures obtained after CHARMM minimization carrying either arginine or glycine at position 69 of the CD33 protein. We did not observe significant changes in the protein backbone after basic energy minimization, but small rearrangements, both in the α-helix containing the glycine residue (red arrow in Figure 1A) and in a β-sheet of the repeated structural type C2 Ig domain (black arrow in Figure 1A) were evident. Most importantly, few differences were observed in the structures of sidechains, which showed small changes in the connecting region between the two domains (Figure 1B).

These predictions led us to hypothesize that the effect of the arginine-to-glycine change at position 69 could predominantly affect the reorganization of polar sidechains, and only locally, the rearrangement of the backbone. In agreement with these observations, the analysis carried out on 200 structural models of CD33 (PDB ID 5IHB), 100 with the Arg69 residue and 100 with the Gly69 residue, indicated the presence of few changes in the boundary spatial coordinates (min and max) of the entire CD33 structure (Figure 2). These differences were hypothesized to be due to the increased length of the sidechain consisting of a three-carbon aliphatic straight chain ending in a guanidino group as a consequence of the change from a glycine to an arginine residue in position 69.

### 2.3. Docking Analysis

The predicted binding sites for sialic acid in the N-terminal domain (aa19-135) of the monomeric form of CD33 (PDB ID 5IHB) carrying either an arginine or a glycine residue in position 69 are reported in Figure 3. The affinity of each binding site toward the ligand is expressed as the binding energy (kcal/mol), with lower values indicating a greater bond affinity. In the Arg69 variant, the values were −4.8 kcal/mol near Arg119 and −5.3 kcal/mol near Arg91, while the values were slightly increased in the form with Gly69, with values of −5.2 kcal/mol near Arg119 and −5.7 kcal/mol near Arg91 (Figure 3).

We found similar results when we performed docking analyses on the other two structures: the N-terminal domain (aa19-135) (PDB ID 5J06, 5J0B), which carried a glycine residue in position 69; and the repeated structural type C2 Ig domain (aa145-228). In the PDB-ID 5J06 structure, the values were −4.7 kcal/mol in Arg119 proximity and −5.2 kcal/mol in Arg91 proximity; while in the PDB-ID 5J0B structure, the predicted binding affinity values were −4.7 kcal/mol in Arg119 proximity and −5.3 kcal/mol in Arg91 proximity (Appendix A). We confirmed these results through the docking analyses carried out on the other three CD33 structures having only the N-terminal domain (PDB ID 6D48, 6D49, and 6D4A). In fact, we always observed a greater binding affinity in the Arg91 residue part than in the Arg119 part of the functional domain (Appendix A), suggesting a better chance of binding at this site. This result was in contrast with the evidence from the binding of a sialic acid mimetic ligand FVP in the proximity of Arg119, as found by others [32]. However, by analyzing the asymmetric unit assembly of each resolved structure in that study, we observed the formation of dimers and tetramers involving interfaces near Arg91, which likely prevented the binding of ligands near this residue. Considering that the most reactive side of the D2:IgV domain is represented by the positively charged patch comprising arginine residues 89 and 91, the oligomerization trend involving this surface could be explained easily. Notwithstanding the site of binding, our results suggested a general increase in the binding affinity and specificity of the glycine variant of CD33 toward ligands.

### 2.4. Virtual Screening Analysis

Plotting the carbon 1 atomic coordinate of the best sialic acid poses against the affinity values (kcal/mol) predicted by the docking analysis, we obtained a conceptual understanding of the ligand distributions on the protein surface with respect to the arginine or glycine variants in position 69 of the CD33 protein. The results highlighted a noteworthy increase in the affinity of several binding sites for sialic acid in the glycine 69 variant of CD33 (Appendix A). Indeed, the analysis of the distributions of the best ligands on the D2:IgV domain of CD33, obtained by plotting the XYZ coordinates of the C1 atom of sialic acid structures (Appendix A), showed binding in specific sites of CD33 receptors for both variants with arginine and glycine residues in position 69. When comparing the position of arginine residues in the D2:IgV domain of the PDB-ID 5IHB structure with the best ligand poses by overlaying the YZ coordinates of the Cζ atom of arginine sidechains with the coordinates of C1 atom of sialic acid poses (Appendix A), we observed a number of sialic-acid-binding molecules close to arginine residues 91, 98, and 119. However, the C1 atom of sialic acid near the approximate Z coordinate 70,000, included in the range of Y coordinates 85,000–95,000 and 110,000–115,000, were instead close to the arginine 62 and 89 residues in the 3D representation, respectively. It was apparent that ligands basically spread out on the surface of the Arg69 structure of CD33, with most of the sialic acid binding near arginine residues 91, 119, 89, 98, and 62, ordered by the number of bonded ligands (Appendix A). In contrast, an increased affinity and number of sialic acid molecules was predicted near the position of arginine 91 in the Gly69 variant of CD33 (Appendix A).

These data suggested the possibility of the presence of more than one binding site for sialic acid derivatives with a similar specificity. In fact, it was not surprising that the more reactive domain near arginine 91 was involved in the binding at the interfaces between the CD33 structures (single and double domains) in the crystallographic unit cell, preventing the recognition of this other sialic-acid-binding site in the crystallographic structures resolved in presence of ligands [32].

The 3D representations in Figure 4 illustrate the best orientations of sialic acid molecules in the CD33 binding sites obtained by virtual screening analysis. In the CD33 structure presenting an arginine residue in position 69, we observed the prevalent distribution of the ligands in the two already identified binding sites involving arginine 91 and 119 (Figure 4A). However, two additional binding sites were predicted near arginine 62 and 89. The number of ligands in these two secondary sites increased in the representation of the second-best poses (Figure 4B), having a lower affinity with respect to the previous ones. In contrast, the best poses of sialic acid molecules were located exclusively near arginine 91 and 119 in the Gly69 variant of CD33 (Figure 4C), and only a few ligand poses were predicted near arginine 62 and 89 (Figure 4D). In addition, when observing the manner the sialic acid binds to the sites on the CD33 surface, we can point out that the orientation of sialic acid molecules was very often preferentially maintained in the binding sites of the Gly69 variant of CD33 (Figure 4). Thus, in addition to the increase in affinity, it was likely that the Gly69 variant of CD33 bound to the sialic acid with a higher specificity than the Arg69 variant. These results supported the idea that the balance of the binding equilibrium constant for CD33–sialic acid interaction changed slightly in a positive direction in the Gly69 variants. This suggested a more efficient binding of sialic acid at a lower concentration by this CD33 variant.

## 3. Discussion

Although the association of CD33 with Alzheimer’s diseases has not been thoroughly elucidated, literature data supports the hypothesis that through the modulation of immune cell functions (such as phagocytosis, cytokine release, apoptosis, etc.), CD33 could be implicated in the suppression of amyloid fragment uptake [12,18,23,25,26,27]. Here, we demonstrated a correlation between an SNP variant of the *CD33* gene (SNP rs2455069-A>G) and Alzheimer’s disease risk, and proposed a new hypothesis for its functional role. Using different in silico approaches, we demonstrated that the binding site for sialic acid could involve additional regions of the protein to the previously documented Arg119 residue. This residue was demonstrated to be essential for the binding of the sialic acid mimetic [32]. Indeed, it was assumed that the binding site for sialic acid was located at the D2:IgV domain of the CD33 receptor, and contained the arginine 119 residue, which was positively charged at physiological pH. Because sialic acid molecules are typically terminal residues on glycoconjugates, the positively charged arginine forms salt bridges with the sialic acid portion of sugar residues, enabling stable interactions [18,33,34,35]. Due to the presence of FVP (a subtype-selective sialic acid mimetic ligand) in the 6D49 and 6D4A structures (recently also demonstrated in the 7AW6 structure), the putative binding site was located near the Arg69 and Arg119 positions (Appendix A). In agreement with those results, we confirmed the binding of sialic acid to Arg119, but also demonstrated the possibility that the ligand may bind to other regions of the protein with higher affinity that include Arg91. The presence of an additional binding site was recently reported for another Siglec family member [36], pointing out the need to further explore the binding properties of CD33 toward sialic-acid-containing molecules. A mutation of arginine 119 to alanine did not affect phagocytosis of U937 cells [27], raising a question on the role of this residue in sialic-acid-containing glycan binding, and supporting a hypothesis of additional binding sites for sialic acid. Our *in-silico* analyses indeed supported the possibility of additional binding sites for sialic acid. In addition, our studies also supported a hypothesis that the CD33-R69G variant may bind sialic acid with greater specificity and affinity, and results in a more efficient binding of the variant to sialic acid, at lower concentrations than that of the wild type. Interestingly, the dependence of binding on sialic acid concentrations was consistent with observations that sialic acid content increases in old-age females [37], who have an increased risk of LOAD compared to males of the same age [38]. In the light of these observations, new perspectives have opened up in the study of CD33, with novel and interesting understandings of the interactions leading to late-onset Alzheimer’s disease. Several polymorphisms in the CD33 gene could affect the structure of this receptor, and in turn, the binding affinity for sialic-acid-containing gangliosides. We hypothesized that the microglial phagocytosis, and in turn, the efficient removal of neurotoxic protein aggregates, might also depend on a variety of factors, including dietary habits, sex-specific accumulation of sialic acid, and aging, among others, affecting ganglioside expression in the brain. This may establish a complex inter-relationship between genetic risk factors and additional predictors in the progression of LOAD. Overall, our results led us to hypothesize that the rs2455069-A>G SNP is a risk factor for LOAD, and that the mechanism of the long-term cumulative effects of the CD33-R69G variant are mediated through an increased binding of sialic acid, which then acts as a more effective enhancer of the CD33 inhibitory effects on amyloid plaque degradation (Figure 5).

## 4. Conclusions

We analyzed the functional role of the SNP rs2455069-A>G, which we found to be associated with Alzheimer’s disease (AD) in a cohort of patients from southern Italy. The SNP changed a single amino acid in the sequence of CD33, and resulted in a local change in its protein structure, which we derived by in silico analysis. Our data demonstrated that this small change resulted in a slight increase in the binding affinity and specificity of CD33 toward sialylated molecules, its natural ligands. We propose that this change in binding enhanced the inhibitory effects of CD33 on amyloid plaque degradation, and together with other genetic, epigenetic, and environmental factors, could predispose to the accumulation of amyloid plaques and the onset of AD pathology. These observations support a new approach to the investigation of protein polymorphisms involved in AD, and propose a need for structural analyses to complements genetic studies.

## 5. Materials and Methods

### 5.1. Participant Recruitment and Analysis

The participants in the study (*n* = 330) were recruited at the Centre for Research and Training in Medicine of Aging (CeRMA) at the University of Molise (Italy). The AD patients (*n* = 195) fulfilled the National Institute of Aging and Alzheimer’s Association (NIA-AA) diagnostic criteria for “probable AD with documented decline” [39]. They scored <24 on the Mini Mental State Examination (MMSE) and >0.5 on the Clinical Dementia Rating (CDR). To rule out other potential causes of cognitive impairment, all patients underwent blood tests (including full blood count, erythrocyte sedimentation rate, blood urea nitrogen and electrolytes, thyroid function, vitamin B12, and folate) and brain imaging. One hundred and thirty-five sex/age-matched cognitively healthy subjects (HS) were recruited as a control group. DNA from all participants was analyzed for the presence of the rs2455069 SNP by high-resolution melting analysis (HRM). Genomic DNA was extracted from whole blood with a QIAamp DNA Blood Mini Kit (QIAGEN). PCR amplifications were performed on a SureCycler 8800 Thermal Cycler (Agilent, Santa Clara, CA, USA). The study was conducted in accordance with ethical principles stated in the Declaration of Helsinki, and with approved national and international guidelines for human research. The Institutional Review Board (IRB) of the University of Molise approved the study (IRB Prot. n. 16/2020). Written informed consent was obtained from each participant or caregiver.

### 5.2. High-Resolution Melting Analysis (HRM)

The high-resolution melting analysis (HRM) was performed as previously described [29] on a LightCycler 480 Instrument II (Roche, Basel, Switzerland) and analyzed using LightCycler^®^ 480 Gene Scanning Software 1.5.1. High-resolution melting Mastermix (Roche) was used. Specific primers used for amplification and sequencing by the Sanger method and the electropherograms were analyzed with Chromas 2.22 software and aligned according to reference sequences present in GenBank (http://www.ncbi.nlm.nih.gov/GenBank/index.html; accessed on 1 September 2019). Templates were amplified in 96-well plates and ~30 ng analyzed in a 20 µL final volume reaction as previously specified [29].

### 5.3. CD33 Structure Editing and Minimization

Computer simulations were carried out on the 3D crystallographic structures available in the Protein Data Bank [31] with ID numbers 5IHB, 5J06, 5J0B, 6D48, 6D49, 6D4A, and 7AW6. The first four included the N-terminal domain (aa19-135) and the repeated structural domain Ig of type C2 (aa145-228), while the last three structures included only the N-terminal domain. All PDB structures of CD33 (excluding 6D48) incorporated ligands. Characteristics of the seven human CD33 structures (Siglec-3) are summarized in Appendix A. The PDB files were edited by using Pymol [40]. Any molecules bound in the protein catalytic sites and water molecules bound to the protein during the crystallization process were removed. The CHARMM-GUI web-based platform (http://www.charmm-gui.org; accessed on 15 February 2022) was used to extract a single monomer from each of the CD33 structures contained in the PDB files, and to generate the two single-nucleotide polymorphisms (SNPs) at position 69 for each monomer. All generated structures were optimized offline using the CHARMM program [41] by performing a basic energy minimization with the steepest descent (SD) algorithm of the initial 25–50 steps to remove bad van der Waals contact. A more precise minimization was carried out subsequently with the Newton–Raphson method (ABNR) of 1,000,000 steps to remove potential problems, such as incorrect contacts, collisions, and nonphysical contacts/interactions. When the step-to-step energy change was less than 0.001 kcal/mol, the structure was considered sufficiently minimized.

### 5.4. Ligand Structure Editing and Minimization

The sialic acid and similar ligand (3′-sialyllactose, 6′-sialyllactose) 3D structures were generated using Avogadro software (http://avogadro.cc; accessed on 15 February 2022), and the structures were optimized through the MMFF94 force field with the steepest descent algorithm until the ΔE reach a value approaching 0 (<10–10).

### 5.5. Docking Analysis

The minimized monomeric structures of CD33 were used for the docking analysis of sialic acid and similar ligand structures. The docking analysis was carried out using Autodock Vina [42]. The ADT software package was used to determine the grid sizes, visualize the ligand poses, and add hydrogen atoms to the templates [43]. We superimposed the different CD33 structures and generated a single grid box that included all the CD33 structures. The coordinates of the grid box were put in a configuration file (config file in Appendix A) used to perform both the docking and the virtual screening analysis. During the docking procedure, both the proteins and ligands were considered as rigid, and the only torsion that was allowed was for the carboxylic and alcoholic groups of the ligand. A bash shell command was used to pass receptors, ligands, and grid box parameters to Autodock Vina, producing an output file containing the predicted models (ligand poses) and a log file for each analysis. The structures were analyzed and the images were produced by using the PyMOL [40] and the UCSF Chimera 1.13.1 (http://www.cgl.ucsf.edu/chimera; accessed on 15 February 2022) [44] molecular graphics software programs.

### 5.6. Virtual Screening Analysis

We prepared two datasets of 100 models each of the Arg69 and Gly69 variants of the CD33 structure (PDB ID 5IHB). We performed a basic energy minimization using the CHARMM program [41], as previously described, to make them more relaxed for subsequent analysis. Using an ad hoc script (see Appendix A) to transfer receptors and grid box parameters for automating the docking process, we submitted the two structure datasets (200 structures total) for analysis by Autodock Vina against the sialic acid as ligand, retrieving at least eight poses for each CD33 structure.

### 5.7. Statistical Analysis

Data were analyzed using the SPSS (v. 17.0) statistical software package (SPSS Inc., Chicago, IL, USA). The frequencies of genotypes and allelotypes of the CD33 gene polymorphism were calculated and determined for deviation from the Hardy–Weinberg equilibrium (HWE). The Chi-squared test and odds ratio (OR) were used to evaluate the association of AD risk with different genotypes and alleles. A *p*-value of <0.05 was considered statistically significant.

## Figures and Tables

**Figure 1 ijms-23-03629-f001:**
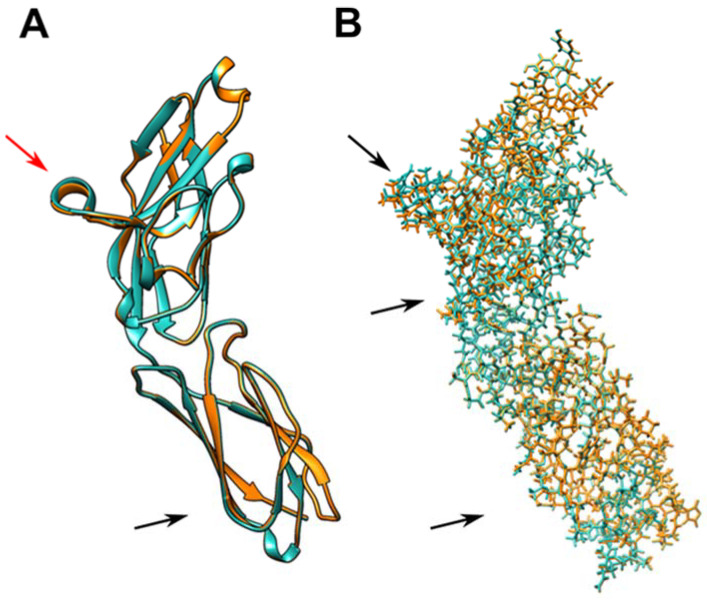
Structural representation of CD33 variants. (**A**) Ribbon representation of the backbone of superimposed CD33 monomers (PDB ID 5IHB) with Arg69 (cyan) and Gly69 (orange) after minimization by CHARMM. (**B**) Stick representation of backbone and sidechains of structures in (**A**). The arrows indicate the area with major changes in the structure of sidechains after basic energy minimization.

**Figure 2 ijms-23-03629-f002:**
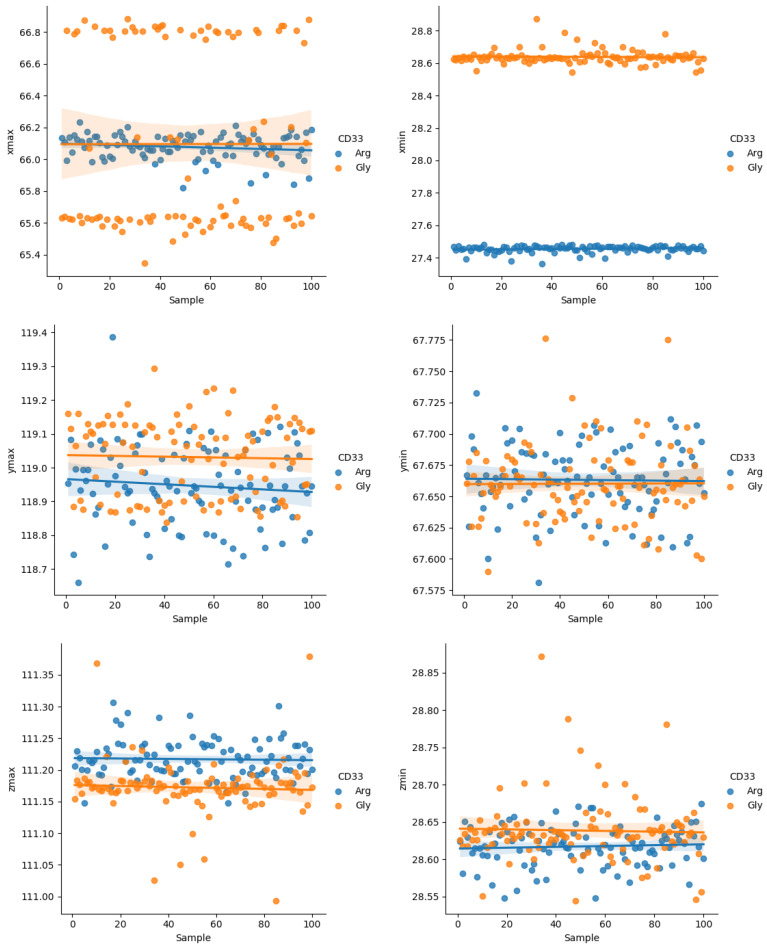
Variations of the maximum and minimum dimensional values in the XYZ axes of the two CD33 variants (PDB ID 5IHB). The variations were obtained after minimization by CHARMM. Blue dots refer to the structures with the arginine residue, whereas the orange dots refer to the structures with a glycine residue in position 69.

**Figure 3 ijms-23-03629-f003:**
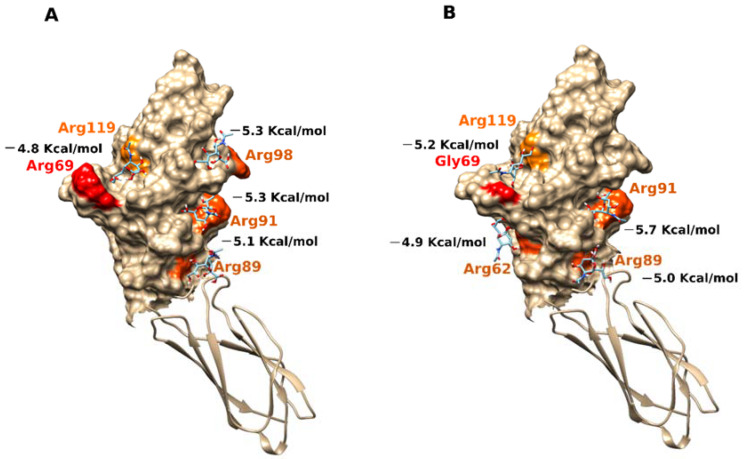
Sialic acid interactions with the two SNPs of CD33 structures. (**A**) Surface representation of functional domain of the CD33 predominant form with Arg in position 69, colored in red (PDB ID 5IHB). (**B**) Surface representation of functional domain of the CD33 variant form with Gly in position 69, colored in red (PDB ID 5IHB). The areas in orange indicate the binding sites of sialic acid, in stick representation, in correspondence with the positively charged arginine residues with which the ligand establishes saline bridges. The values of the binding energy in kcal/mol are reported for each binding site of sialic acid.

**Figure 4 ijms-23-03629-f004:**
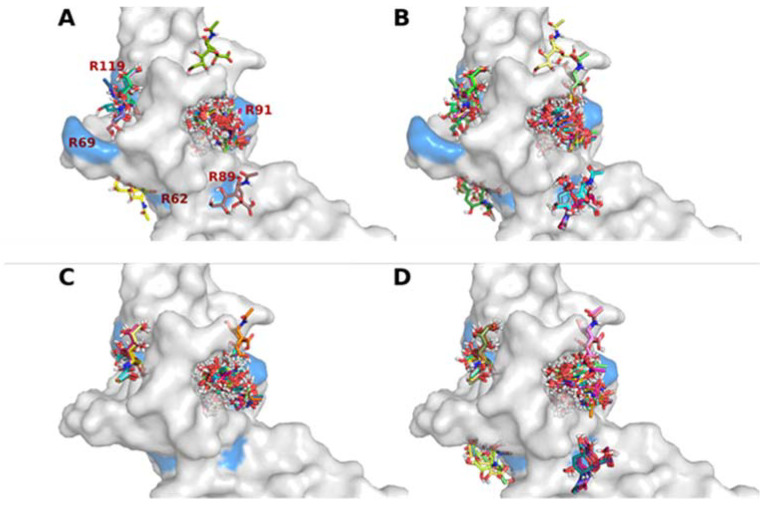
Detail of the binding sites in the D2:IgV domain for sialic acid molecules, as derived from docking analyses. Surface representation, in light gray, of the N-terminal D2:IgV functional domain of arginine 69 models of CD33, in which the docked poses of sialic acid molecules with the best (**A**) and the second best (**B**) ligand mode rmsd are shown in stick figures. Surface representation, in light gray, of the N-terminal D2:IgV functional domain of glycine 69 models of CD33, in which the docked poses of sialic acid molecules with the best (**C**) and the second best (**D**) ligand mode rmsd are shown in stick figures. The surfaces of arginine residues 62, 69, 89, 91, and 119 are presented in light blue.

**Figure 5 ijms-23-03629-f005:**
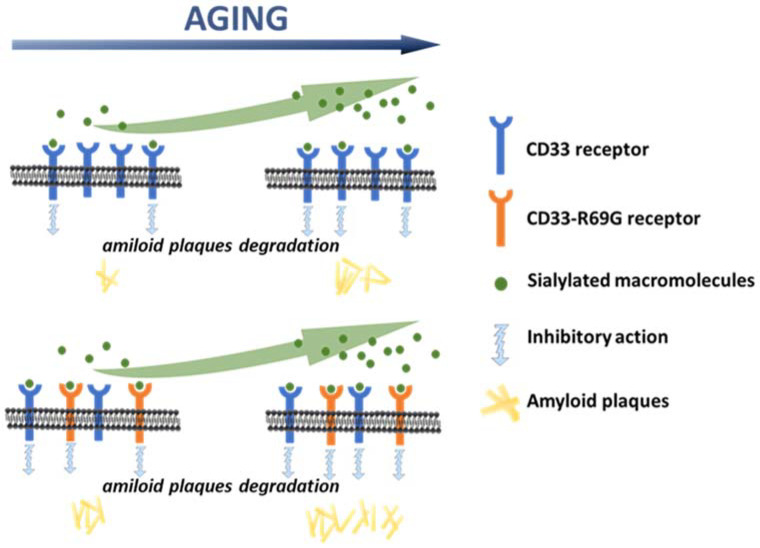
Model describing the plaque accumulation during aging. The increased effect of the CD33-R69G variant in inhibiting the microglial phagocytosis results in a lower extent of clearance of amyloid plaques, supporting late-onset AD.

**Table 1 ijms-23-03629-t001:** Association between the genetic polymorphism CD33-rs2455069 and the risk of AD.

CD33 Genotype	OR ^a^ (95% C.I. ^b^)	Chi2	*p*
AA	Ref		
AG	2.149 (1.259–3.668)	8.02	*p* = 0.00463
GG	3.040 (1.428–6.476)	8.58	*p* = 0.00341
AG + GG	2.294 (1.363–3.862)	10.03	*p* = 0.00154
A	Ref		
G	1.555 (1.135–2.130)	7.61	*p* = 0.00582

^a^ OR = odds ratio; ^b^ C.I. = confidence interval.

## Data Availability

All data are available in the main text or the Appendix A.

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
