# Peer review of "CD33 rs2455069 SNP: Correlation with Alzheimer’s Disease and Hypothesis of Functional Role"

_ijms, 2022, doi:10.3390/ijms23073629_

Round 1

Reviewer 1 Report

Major Comments:

  1. The authors of this study present an interesting follow-up study to their previous report of a potential familial dementia phenotype. The association with CD33 was discovered in a search of “candidate genes including the APP, PSEN1, PSEN2, GRN, TREM2, CD33, MAPT, and APOE genes” (from the author’s publication https://doi.org/10.1093/jnen/nlaa055listed as Reference 13 in this manuscript). The SNP rs2455069-A>G, a missense change is in LD with 10 other SNPs on Exon 2. In the 1000 genomes data the population frequency of this variant in the group identified as “Toscani in Italy” is 47.2%. Can the authors discuss the implications of their results in this context?
  2. Can they specifically discuss their hypothesis of familial FTD and AD that originally led to CD33 and now its description in LOAD?
  3. Is the genotyping data of the controls and cases available on any database as cautionary notes for genotype frequency not being in HWE have been suggested previously https://www.frontiersin.org/articles/10.3389/fgene.2017.00167/full? Alternatively, is there any way to confirm the accuracy of genotyping?
  4. This was perhaps not tested and may have been outside the scope of the manuscript. The authors can say so. Was the status of other polymorphisms such as those that reduce CD33 expression were also evaluated in these subjects?
  5. This reviewer does not have the expertise to comment on the in silico analyses and interpretation based on those analysis, so I differ this to the other reviewers.

Minor Comments:

Please go through copy-editing to correct a number of spelling errors. Some examples below.

Line 34/35: Provide reference for this estimate.

Line 36/37: The cited reference links to “World Alzheimer Report 2019 Attitudes to dementia”. Is an attitude suvey the basis of the author’s introduction that “AD is considered the most common form of dementia accounting for 50–75% of 35 cases”. The problem with this kind of short hand is that it inflates figures of incidence and can undermine analysis of global burden of diseases. Please cite appropriate scientifically validated estimates.

Line 38: Please cite how the number of 5 to 10% was determined?

Line 38: 5-10% are “familiar” correct to “familial”

Line 38: Authors state “GWAS identified more than 20 genetic loci, including CD33”. Can they confirm that CD33 was discovered as one of the GWAS LOAD loci in the refences 2 as well? Nat Genet 51, 414–430 (2019) quoted as Reference 2 quotes an earlier meta-analysis reference from 2011 Nat Genet 2011 May;43(5):436-41. Please choose the most appropriate ref.s as some meta-analyses haven’t found CD33 association.

43 to 46: Please rephrase, or break into two sentences, as it is confusing, and the nuance of the speculation based on the reduction of CD33 expression and ABeta- clearance (in Ref. 8) may be lost.

Line 56: Please rephrase. The presence of the “minor allele” at rs2455069 instead of “The presence of this SNP”, as I suppose this is what the authors mean.

Line 62/244: Change “medicated” to “mediated”

Line 72: “and 16.9” to and 16.9%

  1. Was binding to P22 predicted in this study?
  2. A more elaborate introduction will be helpful for readers to place the study in the context of other studies.

Author Response

Major Comments:

We thank the reviewer for his effort and suggestions that allowed us to clarify the study and improve its presentation. Below, we have tried to address each comment with a specific, clarifying discussion.

  1. The authors of this study present an interesting follow-up study to their previous report of a potential familial dementia phenotype. The association with CD33 was discovered in a search of “candidate genes including the APP, PSEN1, PSEN2, GRN, TREM2, CD33, MAPT, and APOE genes” (from the author’s publication https://doi.org/10.1093/jnen/nlaa055listed as Reference 13 in this manuscript). The SNP rs2455069-A>G, a missense change is in LD with 10 other SNPs on Exon 2. In the 1000 genomes data the population frequency of this variant in the group identified as “Toscani in Italy” is 47.2%. Can the authors discuss the implications of their results in this context?

R1:  To clarify the context of our data, we focus attention to several relevant allelic frequency distributions. The G allele frequency in a pooled European population from 1000 genomes varies from 41.4% in Finnish and 43.9% in Iberian individuals from Spain to 47.2% in individuals from Tuscany, Italy. In contrast, the gnomAD v2.1.1 database reported to G allele frequency to be 41.7% in non-Finnish Europeans controls, which is very similar to that found in our controls (39.6%). In addition, a relatively divergent genetic structure between the south and the north of Italy is well documented, with Tuscany being closer to the North than to central or southern Italy (Raveane A et al., Science Advances 2019, Sazzini M et al., BMC Biology 2020). In light of these data, the frequencies found in our study may be explained by their well-established Southern Italy geographic origin, found to be the same for both cases and controls. Therefore, we are quite confident that the difference between cases and controls is a real effect.

  1. Can they specifically discuss their hypothesis of familial FTD and AD that originally led to CD33 and now its description in LOAD?

R2: Dementia represents a heterogeneous group of neurodegenerative disorders characterized by impairment of the most important cognitive and behavioral functions, and a loss of independence in daily activities. Dementia is not a single disease, so when we reference dementia, we implicate a wide range of specific medical conditions, including AD. Dementia leads to death as a common path in all its forms. Both Alzheimer’s disease (AD) and Frontotemporal dementia (FTD) are heterogeneous with respect to clinical features and underlying pathologies. They have overlapping clinical symptoms, are multifactorial, are associated with many candidate genes, and probably involve common molecular pathways that are yet to be understood. It is common for people with dementia to have more than one form of dementia. Mixed dementia phenomena exist, and they initiate and influence each other. Further knowledge gains in this area will help researchers to better understand these conditions and develop more personalized prevention and treatment strategies. We identified a tag SNP in CD33 exon 2, rs2455069-A>G, as part of an LD block associated with the disease in a single-family study, and the association was confirmed in LOAD patients of the cohort analyzed in this study. The block crosses the IgSF cluster region and has a potential regulatory function on gene expression of different Siglec’s member isoforms. In fact, CD33 and SIGLECL1 exhibit significantly increased levels of expression in LOAD and, to a lower extent, in the H217 patient (Rendina et al., 2020).

  1. Is the genotyping data of the controls and cases available on any database as cautionary notes for genotype frequency not being in HWE have been suggested previously https://www.frontiersin.org/articles/10.3389/fgene.2017.00167/full? Alternatively, is there any way to confirm the accuracy of genotyping?

R3: We thank the reviewer for having raised this point and for citing a relevant paper on the Hardy-Weinberg Equilibrium (Chen B et al., Front Genet 2017). We have now calculated HWE separately in AD patients and in controls. Indeed, we found that affected cases deviated from the equilibrium, but controls did not have a statistically significant deviation in their distribution (p=0.083). While the number of heterozygous individuals in our controls is slightly higher than expected, this is in line with a genotyping error suggested in the paper proposed by the reviewer (Chen et a., Front Genet, 2017). Notwithstanding, the deviation was not statistically significant. In addition, we confirmed our analysis using both the High-Resolution Melting Analysis (HRM) technique and the direct Sanger sequencing method for more than half of the samples randomly chosen. We are therefore highly confident that our genotyping results are accurate. The changes are incorporated in the manuscript text.

  1. This was perhaps not tested and may have been outside the scope of the manuscript. The authors can say so. Was the status of other polymorphisms such as those that reduce CD33 expression were also evaluated in these subjects?

R4: This was not tested, and it is indeed outside the scope of the manuscript.

  1. This reviewer does not have the expertise to comment on the in silico analyses and interpretation based on those analyses, so I differ this to the other reviewers.

Minor Comments:

Please go through copy-editing to correct a number of spelling errors. Some examples below.

R: In agreement with the reviewer suggestions, we reviewed the copy-editing and corrected any spelling, grammatical and syntax errors.

Line 34/35: Provide reference for this estimate.

R: In agreement with the reviewer suggestions, we changed the sentence regarding the estimation of the future dementia cases of and added two references (1 and 2).

Line 36/37: The cited reference links to “World Alzheimer Report 2019 Attitudes to dementia”. Is an attitude survey the basis of the author’s introduction that “AD is considered the most common form of dementia accounting for 50–75% of 35 cases”. The problem with this that it inflates figures of incidence and can undermine analysis of global burden of diseases. Please cite appropriate scientifically validated estimates.

R: In accordance with the reviewer comment, we changed the sentence as follow “with AD being the most common form of dementia currently [3,4].”

Line 38: Please cite how the number of 5 to 10% was determined?

R: In accordance with the reviewer comment, we corrected the percentage of EOAD based on the estimation reported in the paper to the number 5.

Line 38: 5-10% are “familiar” correct to “familial”

R: In accordance with the reviewer comment, we corrected typing error.

Line 38: Authors state “GWAS identified more than 20 genetic loci, including CD33”. Can they confirm that CD33 was discovered as one of the GWAS LOAD loci in the refences 2 as well? Nat Genet 51, 414–430 (2019) quoted as Reference 2 quotes an earlier meta-analysis reference from 2011 Nat Genet 2011 May;43(5):436-41. Please choose the most appropriate ref.s as some meta-analyses haven’t found CD33 association.

R: We corrected the sentence as suggested by the reviewer.

Line 62/244: Change “medicated” to “mediated”

R: We corrected the sentence as suggested by the reviewer.

Line 72: “and 16.9” to and 16.9%

R: We corrected the sentence as suggested by the reviewer.

Was binding to P22 predicted in this study?

R: We would thank the reviewer to focus on this point, because we wrongly reported the name of sialic acid mimetic compound (P22) as used by the cited authors. the molecule is the FVP in all the pdb files. Thus, for clarity we removed the P22 name from the paragraph.

Anyway, we already did a prediction of the FVP to the CD33 structures in which we added the arginine residue, and we obtained similar results for both side of the molecule, involving respectively the arginine 119 and 91, in particular, in the latter site (near arginine 91), we obtained 5 poses ranging from -7.6 to -8.0 kcal/mol with respect only 3 poses, in the same range of affinity values, near the arginine 119. Unfortunately, we cannot stress this result, because the 3D structures solved in presence of the FVP maintained a better binding pocket near the arginine 119 (being co-crystallized), and a more extreme minimization of the structures, in order to remove this three-dimensional constrain, could result in an artifact. Therefore, these data appear to us to be too speculative and not suitable for use in the paper.

A more elaborate introduction will be helpful for readers to place the study in the context of other studies.

R: we thank the reviewer for the suggestion. In this regard, we extended the introduction section adding new information and references.

Reviewer 2 Report

Reading and studying the manuscript made by Tortora et al. it was very interesting. In this paper, the authors aim to find correlations between the polymorphisms of the CD33 gene, a gene that encodes a member of lectin and binds sialic acid. It is known that many of these polymorphisms being associated with an increased risk of A.D. The present study is all the more valuable as it also includes a clinical part. The authors were able to show that there is a correlation between CD33 SNP rs2455069 and Alzheimer's Dementia in patients in a region of southern Italy and, secondly, that the change in amino acids R69G affects the structure of CD33 in a way that alters the binding affinity for sialic acid residues, which acts as an enhancer of the inhibitory effects of CD33 on amyloid plaque degradation.

The article is written in a concise and orderly manner, respecting the structure of the journal. However, I have a few minor comments:

  1. The English language, although quite good, should be improved.
  2. In the discussion section related to the clinical trial conducted in southern Italy, I would like to add other relevant clinical trials conducted in different geographical areas of the globe.
  3. Although the conclusions are not binding, it would still be more appropriate if they were mentioned at the end of the article, making it easier to read your research.
  4. You could improve the bibliographic references by adding recent studies (2020, 2021), thus strengthening your study and interest in this topic.

Author Response

We thank the reviewer for his effort and suggestions that allowed us to clarify the study and improve its presentation. Below, we have tried to address each comment with a specific, clarifying discussion.

Reading and studying the manuscript made by Tortora et al. it was very interesting. In this paper, the authors aim to find correlations between the polymorphisms of the CD33 gene, a gene that encodes a member of lectin and binds sialic acid. It is known that many of these polymorphisms being associated with an increased risk of A.D. The present study is all the more valuable as it also includes a clinical part. The authors were able to show that there is a correlation between CD33 SNP rs2455069 and Alzheimer's Dementia in patients in a region of southern Italy and, secondly, that the change in amino acids R69G affects the structure of CD33 in a way that alters the binding affinity for sialic acid residues, which acts as an enhancer of the inhibitory effects of CD33 on amyloid plaque degradation.

The article is written in a concise and orderly manner, respecting the structure of the journal. However, I have a few minor comments:

  1. The English language, although quite good, should be improved.

R1: we extensively review English language as suggested by the reviewer.

  1. In the discussion section related to the clinical trial conducted in southern Italy, I would like to add other relevant clinical trials conducted in different geographical areas of the globe.

R2: As also replied to the review 1 in R2, we identified a tag SNP in CD33 exon 2, rs2455069-A>G, as part of an LD block associated with the disease in a single-family study, and the association was confirmed in our cohort of LOAD patients from Southern of Italy analyzed and presented in this study. Clinical trials were not done and are indeed outside the scope of the manuscript.

  1. Although the conclusions are not binding, it would still be more appropriate if they were mentioned at the end of the article, making it easier to read your research.

R3: In agreement with the reviewer suggestion, we added a conclusions section.

  1. You could improve the bibliographic references by adding recent studies (2020, 2021), thus strengthening your study and interest in this topic.

R4: we extended the introduction section and added new recent references as suggested by the reviewer.